# Reputation as Capital—How Decentralized Autonomous Organizations Address Shortcomings in the Venture Capital Market

## Wulf Kaal

School of Law, University of St. Thomas, Minneapolis, MN 55403, USA; wulf@wulfkaal.com

**Abstract:** Venture capital (VC) models can be optimized with emerging decentralized technology. There are many disadvantages that come with traditional VC fundraising including illiquidity and ownership struggles, as well as timing. This paper will discuss alternative funding mechanisms that may be available and advantageous to emerging businesses. After discussing the shortcomings of the existing VC market and the rise of alternative early round funding mechanisms, the paper highlights the evolution of VC businesses that are operated by a Decentralized Autonomous Organization (DAO). More specifically, models discussed in this article contribute to the much-needed experimentation with venture capital reputation models.

**Keywords:** venture capital; Decentralized Autonomous Organization; reputation; decentralized governance; capital; venture funding; finance; token models; cryptocurrencies; feedback effects; emerging technology; tokens; blockchain; distributed ledger technology

## 1. Introduction

Financing options available to technology entrepreneurs are increasing. The hitherto almost exclusive focus on the access to venture capital (VC) investing firms is slowly eroded through a broadening spectrum of funding opportunities for technology startups. The spectrum of funding opportunities has expanded with the proliferation of independent business angels (BAs), business angel networks (BANs), crowdfunding platforms, initial coin offerings (ICOs), and initial decentralized exchange offerings (IDO), among others. These emerging and expanding decentralized funding opportunities are increasingly utilized, especially in the Silicon Valley tech startup industry.

Digital asset startups that grow rapidly into billion-dollar businesses with little or no VC funding have demonstrated that the traditional venture capital model alone was not enough to meet the funding needs of technology startups. As one case in point, in the late 2010s, Silicon Valley VC firm Andreesen Horowitz recently renounced its SEC VC classification and exemptions in favor of registering as an investment advisor company (Konrad 2019). This move was motivated largely by a desire to give Andreesen Horowitz enhanced flexibility in investing in blockchain-oriented financing strategies among other newer, riskier, and more unconventional capital investment methods with higher payout opportunities. Other VC firms have joined Andreesen Horowitz in moving away from conventional VC endeavors towards a more modern startup investment approach tailored to the emerging blockchain and crowdfunding technologies of the 21st century.

The current state of the VC market is trending towards a combination in the types of fundraising, starting with ICO and crowdfunding platforms to raise an initial round of capital, generate interest and publicity for the startup, and act as a litmus test for pitching the startup to BAs and VCs once there has been some success with the early rounds. Nevertheless, despite all the success stories, there is a low rate of investor success

with all four varieties of capital investment (see ibid.; see also Schwienbacher 2019, p. 65; Bonini et al. 2019, pp. 13–49).

This article shows that decentralized reputation governance models in venture capital have the potential to upgrade the venture capital market. After discussing the shortcomings of the legacy VC market and emerging models for early round funding of digital asset startups, the paper evaluates reputation as capital as another alternative of the early round funding methods. The paper highlights the model evolution of reputation as capital models.

## 2. Shortcomings of the Existing VC Model

The existing VC model is subject to significant ongoing downsides that inhibit its ability to provide long-term value to the tech startup community. The existing market for fiat VC investments is subject to several significant downsides, especially in comparison with the market for early round digital asset investments.

### 2.1. Fiat VC Market

2.1.1. Liquidity—Capital Calls and Raising Capital

One of the biggest problems in the traditional VC ecosystem is lack of liquidity (Eha 2017). The traditional VC model disincentivizes generating early profits (VNX Exchange 2019, p. 5). Capital is "locked in" or "locked up" for an extended period of time. Once an investment is made, its success is highly dependent on a small group of managers and/or entrepreneurs (Fried and Hisrich 1994). Investing in innovative technology requires substantial time to generate returns—at least seven years—and requires even longer to generate a profitable exit (usually via an IPO or M&A transaction) (VNX Exchange 2019, p. 4). VC firms collect capital from Limited Partner investors for a period of 8–12 years (VNX Exchange 2019, p. 4). Venture capitalists invest with a view towards capital gain on exit and overall liquidity (Cumming and Johan 2012). In order to maximize liquidity, the most preferred routes are the most profitable exit routes (ibid., p. 3). Moreover, VC investing is high-risk and subject to regulation that negatively impacts liquidity.[1] Due to these liquidity downsides, companies are increasingly moving toward a "staying private" strategy by decreasing venture-backed exits (VNX Exchange 2019, p. 5). Early-stage funding fell from 35 to 27% of total VC funding as VCs increasingly focus on funding later-stage companies (VNX Exchange 2019, p. 6).

The existing market for fiat VC requires the fiat VC funds to support capital calls with liquidity (Ang and Sorensen 2012, pp. 1–56). This means that VCs cannot deploy all the capital they raised in order to attain higher returns as they may be subject to capital calls that need to be funded. Due to their ability in maintaining liquidity to support capital calls from their investors, traditional fiat VC funds are limited in their ability to deploy capital. Similar to insurance companies that are required to maintain certain cash reserves to fund possible policy payouts, traditional fiat VC funds need to maintain assets to fund future liabilities which affects their return on investment and overall fund performance (Siegel 2008).

Traditional fiat VCs are also required to spend a significant amount of their time raising capital from limited partners. Consequently, some argue that new sources of capital raising can reduce the time and cost for entrepreneurial ventures to raise funds (Butticè and Vismara 2022, pp. 1224–41). This time commitment to raising capital often distracts VCs from their main investing responsibilities as they are forced to spend more time with potential limited partners and their needs rather than dealing with due diligence on potential new and/or existing portfolio companies (Finsmes 2019). In other words, time spent on limited partners and their needs is time taken away from the portfolio company due to diligence and efforts to make such portfolio companies successful. Due to this dynamic, the current VC model tends to reflect the networking and public relations ability of the VC's general partner instead of their capital allocation ability (Manigart 2005).

### 2.1.2. Downsides of Syndication

The VC industry started small (Bruton et al. 2005), so in order to diversify risk while maximizing return, firms syndicated their investments with other firms (Bruton et al. 2005, p. 739; citing Reiner 1989). Even as firm size grew and the need to syndicate in order to diversify risk declined, firms continued to syndicate in order to share resources including knowledge, legitimacy, and deal flow (Bruton et al. 2005, p. 739; citing Sorenson and Stuart 2001, p. 1546). Trust is central to the success of syndication (Bruton et al. 2005, p. 739; citing Sorenson and Stuart 2001).

Syndication in traditional VC investment can be efficient and create economies of scale because VCs serve as an information-producing agent and decision-making agent for the investor.[2] VCs actively search for prospects via their informal network, conference attendance, and executive search agencies in an identified market (Tyebjee and Bruno 1984, pp. 1051, 1055–56). Even in 1994, a VC had lower information-gathering costs through economies of scale, scope, and a learning curve (Fried and Hisrich 1994, pp. 35–36). The role of an intermediary is valuable, but in 1994, VC funds traditionally paid a management fee of 2.5% of assets per year, as well as 20% of any profits (Fried and Hisrich 1994, p. 36). A field study found firms from three US regions shared the same "dominant logic" or sets of beliefs about how VC firms and their portfolio companies should behave (Bruton et al. 2005, p. 739; citing Fried and Hisrich 1995, p. 101). This is unsurprising because professional standards with a strong normative set of beliefs for industry practice are set by a centralized entity—The National Venture Capital Association. Its membership is strictly voluntary with expensive membership fees.

Yet, syndication and deal organization is time consuming and expensive for traditional VCs. Deal organization refers to the processes by which deals enter into consideration as investment prospects (Tyebjee and Bruno 1984). Intermediaries play an important role in finding VC prospects (Tyebjee and Bruno 1984, p. 1052) because the typical investment prospect is too small a company to be readily identifiable as a potential candidate (Tyebjee and Bruno 1984, p. 1052). Intermediation drives up cost. Referrals are made through syndication, prior investees and personal acquaintances, banks, and investment brokers. These referrals are more likely to pass through the first screen because the capitalist has confidence in the referrer's judgment and the referrer is more likely to understand what type of investments may appeal to the capitalist (Fried and Hisrich 1994, p. 32). Syndication is a referral within the VC community where the referrer is the lead investor seeking participation of other VC funds.[3]

### 2.1.3. Cost of Deal Screening and Structuring

Deal screening is a significant cost factor for traditional VC firms. Firms with small staffs screen a relatively large number of potential available deals (Tyebjee and Bruno 1984, p. 1052). Investments are typically limited to areas with which the VC is familiar, particularly in terms of the technology, product, and market scope of the venture (Tyebjee and Bruno 1984, pp. 1052–53). Initial screening is based on criteria of (1) investment size and policy of the fund; (2) venture technology and market sector; (3) venture geographic location, and (4) financing stage.[4]

Deal evaluation and risk assessment in a traditional VC is fraught with inaccuracies and suboptimal incentives. VCs weigh several characteristics to assess the perceived risk and expected return of a prospective venture in deciding whether or not to invest (Tyebjee and Bruno 1984, p. 1051). VCs subjectively assess the prospect's business plan because the majority of ventures in search of capital have little, if any, operating history (Tyebjee and Bruno 1984, p. 1053). The analysis is multi-dimensional (Tyebjee and Bruno 1984, p. 1053), in the areas of (1) market attractiveness, (2) product differentiation, (3) managerial capabilities, (4) environmental threat resistance, and (5) cash-out potential (Tyebjee and Bruno 1984, pp. 1051, 1053–54). A potential drawback of this process is that perceptual, emotional, and cognitive processes impact the decision in addition to financial decisions because the evaluation criteria is made through subjective human decision-making (Fried

and Hisrich 1994, p. 29). Additionally, none of those criteria reflect how a prospective deal may correlate with one already in the capitalist's investment portfolio (Tyebjee and Bruno 1984, p. 1053).

Finally, significant information asymmetries can lead managers to engage in opportunistic behavior after an investment is made (Fried and Hisrich 1994, p. 28). Venture capitalists are not involved in day-to-day operations of the venture, but they can intervene in the case of a financial or managerial crisis, and because capitalists typically want to cash out their gains five to ten years after initial investments, they play an active role in directing the company towards a merger, acquisition, or public offering.[5] This can have significant downsides for the portfolio companies and their products.

### 2.2. Digital Asset VC Market

Traditional VCs often experience significant problems in entering the market for digital assets or maintaining their momentum in the market. Several factors help explain their struggles. Among those are the significant volatility of the digital asset market and lacking expertise of traditional VCs in the digital asset market which is still dominated by several key specialized players (Lin and Nestarcova 2019). Moreover, traditional VCs often cannot effectively compete with the ever-increasing array of decentralized token offering avenues.

Traditional VCs are often also not able to effectively invest in the digital asset market because they need to defend their investment choices to their own investors (Zeider 1998). In general, crypto-assets require non-traditional methods of valuation in order to gauge their success, namely, due to a lack of cash flow (Lin and Nestarcova 2019, p. 18), and because of that, traditional VCs are often reluctant to invest in digital asset startups with little history or sales records. Digital asset startups often increase the level of due diligence for VCs because digital asset startups typically have less trackable information on product and performance (Bonini and Capizzi 2019; Löher 2016; Hornuf et al. 2018; Konrad 2019). At the same time, successful digital asset startups provide a significant upside for VC investors. As a result, some traditional VCs create a digital asset arm or specialize entirely.

Additionally, SEC rule 203(l)-1(a) poses another barrier for traditional VC investment into digital asset ventures. Rule 203(l)-1(a) requires that "[i]mmediately after the acquisition of any asset, other than qualifying investments or short-term holdings, [hold] no more than 20% percent of the amount of the fund's aggregate capital contributions and uncalled committed capital in assets (other than short-term holdings) that are not qualifying investments, valued at cost or fair value, consistently applied by the fund."[6] Therefore, because many digital assets are considered liquid when listed on an exchange, a VC is not entitled to hold more than 20% of their capital assets that could be considered as liquid (Cardenas 2018). Within the crypto-community, there is no single accepted method of valuation considering the impressive range of volatility present in different digital assets/currencies.[7] Without a reliable valuation method that can be used, traditional VC firms are hesitant to invest in digital assets that may push their portfolio over the current 20% constraint when holding liquid assets.

### 3. Alternative Early Round Funding Mechanisms

Alternative early round funding mechanisms fill an important gap in the funding options available to tech entrepreneurs. VCs are typically looking for the next Facebook. However, the lack of a company track record makes investing in such companies informed speculation at best (Konrad 2019). As a rule of thumb in the early investment rounds, the lower the information asymmetry, the lower the payout. Below a certain threshold of established value, VCs will rarely invest. In order for tech entrepreneurs to reach that threshold, alternative early round funding options, BAs, BANs, ICOs, IDOs, and crowd-funding platforms are becoming increasingly important (Lin and Nestarcova 2019). These alternative funding methods are important to get entrepreneurs over the VC investment threshold, or to help entrepreneurs completely bypass the VC model (Bonini and Capizzi 2019).

At its peak in the 2019 cycle, the volume of ICOs has surpassed VCs and BAs as a fundraising method for startups (Lin and Nestarcova 2019). BAs are particularly attractive to tech startups in comparison with the VC market (Bonini et al. 2019; Löher 2016) and VCs are increasingly taking a backseat to newer forms of early round startup funding. Furthermore,

> Tech entrepreneurs are increasingly drawn towards the less formal relationships that are typically cultivated between alternative early round funding methods and tech entrepreneurs. Whereas strict contractually-focused relationships with VCs are focused primarily on a data-driven approach that emphasizes financial returns, putting less emphasis on helping with the growth of the startup, alternative early round funding mechanisms often develop a more personal mentor–mentee relationship with the entrepreneur (D'Ambrosio and Gianfrate 2016). Alternative funding methods are particularly beneficial as they are able to be tailored to specific projects and their needs rather than a generalized profit-centric VC model (CB Insights 2021). Such alternative early round funding mechanisms often provide effective advice and assistance in formulating a business plan and marketing the startup company (Bonini and Capizzi 2019). This soft monitoring approach is increasingly favored over the hard monitoring conducted by VCs (Bonini and Capizzi 2019). Key for the success of the relationship-based approach of alternative funding mechanisms is often the establishment of an effective developer team that can help the technology come to fruition.

Key distinguishing features often favor alternative early round funding mechanisms over traditional VC investments. Information asymmetries increase in traditional VC investment rounds as startups and are incentivized to self-censor when engaging with VCs, as they have little data to work with, and are reluctant to overexpose themselves to prospective investors (Bonini and Capizzi 2019). By contrast, the soft monitoring approach of alternative early round funding mechanisms tends towards a nurturing, mentoring role which not only encourages forthrightness, but actually coaches these startups on how to present themselves to other investors in further fundraising rounds, and how to market the company to the public.

Traditional VC fee-based compensation mechanisms are often less competitive than the compensation structures used by alternative early round funding mechanisms. The typical VC fee-based compensation structure can lead to serious shortcomings, including excessive fundraising, suboptimal investments, misevaluation, and overfunding of the portfolio companies during a fund's holding period. Technologically sophisticated tech entrepreneurs often prefer alternative early round funding mechanisms in the first financing round, whereas VCs are more capable of longer-term value enhancement (Löher 2016). Due to this dynamic, tech entrepreneurs are required to engage in a daunting balancing act between alternative early round funding mechanisms and VCs, and in deliberating between the soft monitoring benefits provided by alternative early round funding mechanisms and the value-added benefits of VCs.

Alternative early round funding mechanisms are not necessarily more successful than VCs in early round tech investing. Whereas alternative early round funding mechanisms often tend to act out of intuition and personal experience, VCs typically use a primarily data-driven approach to investment decisions and portfolio company evaluation and supervision. Tech entrepreneurs who work with alternative early round funding mechanisms are subject to external factors affecting such early round investors individually, including liquidity problems, eccentricity, personality issues, and differing visions for management of the company's present and future. In general, equity crowdfunding presents an issue with liquidity in secondary markets making them more challenging to exit for investors (Butticè and Vismara 2022, pp. 1224–41). These shortcomings can contribute to the success but also to the downfall of alternative early round funding mechanisms.[8]

The evidence on the success and failure rate of alternative early round funding mechanisms is mixed. These trends are continuously evolving. The analysis of crowdfunding and

ICOs, among other alternative early round funding mechanisms, is ambiguous at best as little is known about the investors themselves in these contexts (Löher 2016; Schwienbacher 2019). The evidence suggests that startups are generally not relying on VCs for capital until later rounds of fundraising, and use alternative early round funding mechanisms for the earlier rounds (Bonini and Capizzi 2019; Konrad 2019; Lin and Nestarcova 2019). Overall, the volume tech startup relationships with alternative early round funding mechanisms has grown to match that of VC-startup relationships (Bonini and Capizzi 2019). While alternative early round funding mechanisms are increasingly used by tech startups in early rounds (Lin and Nestarcova 2019; Bonini and Capizzi 2019; Hornuf et al. 2018; Schwienbacher 2019), these methods frequently fail to provide adequate funding to fully launch a new company. A significant number of tech startups that rely exclusively on these methods fail (Schwienbacher 2019; Konrad 2019).

## 4. Reputation as Capital

The concept of reputation as capital introduces another early round funding alternative. The concept introduces several optimization metrics compared with other early round funding alternatives.[9]

The novelty of reputation as capital lies in the decentralized and democratized capital allocation functionality enabled by the DAO design. The DAO proposed herein consists of a group of VCs, who roughly correspond to the general partners or shareholders of a venture firm. These VCs each hold a predefined number of reputation tokens in the DAO which allows them to stake their reputation on certain deal/investments that are listed on the non-custodial deal platform of the DAO. The VC's proportional holdings of these reputation tokens is likely to increase over time if the VC follows sound and successful practices by way of staking the VC's reputation tokens on investment proposals and succeeding in the selection of portfolio companies together with other VCs who also stake their reputation tokens for or against a given investment proposal.

Reputation tokens serve multiple purposes in the VC investment process. It is important to note that the reputation tokens are separate and distinct from the fiat currency or other fungible tokens that may be used to pay for investments in portfolio companies.

First and foremost, only reputation token holders are allowed to participate in the portfolio selection process which materializes by way of reputation staking on investment proposals.

Second, reputation tokens serve as claims on the future cash flows generated by the DAO. These cash flows in the form of fungible cryptocurrencies are paid in proportion to each DAO members' non-fungible reputation token holding. More specifically, the returns on investments (ROI) collected by the DAO are distributed among the reputation token holders in proportion to their reputation token holdings. The ROI functions as a reputation salary that gets paid in proportion to each VC's reputation token holdings. It is important to note that ROI is only partially considered the revenue of the individual VCs who committed capital. Rather, the ROI is treated as DAO revenue that gets partially shared pro-rata based on investment amounts committed by each VC and partially shared as a reputation salary which is shared among the DAO's VC participants. Accordingly, the value of the reputation tokens is a function of the ROI of the DAO.

Furthermore, VCs who support investment proposals by staking their reputation tokens are rewarded with a certain number of newly minted reputation tokens as salaries, based on a predetermined formula. Accordingly, the total number of tokens grows over time.

The reputation as capital model emphasizes the GP's reputation in capital allocation which would, over time, free the GPs from having to commit actual capital if they stake their reputation. If VC reputation is truly meaningful, whales can commit/stake reputation to fund a project that is then not actually funded with capital by the whale but rather by the market/LPs. If whales stake but the market commits for their stake with capital, whales should still have the right to receive (part of) the ROI because it is their reputation that

is at stake. Whales are not required to commit capital and do not have to support capital calls with liquidity. Networking and PR would take place at the beginning of the Fund through the Founders—e.g., 'ride with the whales' but as whale reputation solidifies, they no longer have to commit capital to deals they staked reputation for. The system with the best incentive alignment and therefore with the most long-term likelihood of success emphasizes the reputation of its members as a means of capital allocation in a fair and equitable way.

## 5. Model Evolution

Calcaterra, Kaal, and Rao modeled the replacement of capital in the context of underwriting (Calcaterra et al. 2019). Reputation as venture capital is different and has to be distinguished from the reputation as capital replacement that is enabled in Calcaterra et al.'s model. Unlike decentralized underwriting where capital is only needed when a file is claimed, and premia from the underwriting contracts can continue to fund those who stake reputation on an underwriting engagement in lieu of capital, reputation in venture capital needs to fill the role of capital for each funded venture deal.

In the decentralized underwriting model, premia are paid by customers on the underwritten insurance policies and subsequently fund the reputation staked to underwrite policies. The market only replaces reputation as capital if and when a reputation underwriter decides not to stake or to otherwise sell the reputation to the open market. In that case, the market will jump at the opportunity to replace a reputation holder in the underwriting DAO because market participants are looking to participate in the flow of premia that are used as reputation salaries. By contrast, using the reputation as a venture capital model, the reputation is funded over time by the market for each venture deal for which venture capital funds are staking reputation tokens in addition to capital to fund the respective deal (Calcaterra and Kaal 2021, p. 184). Figures 1–4 are original works made for the purpose of this article as shown below.

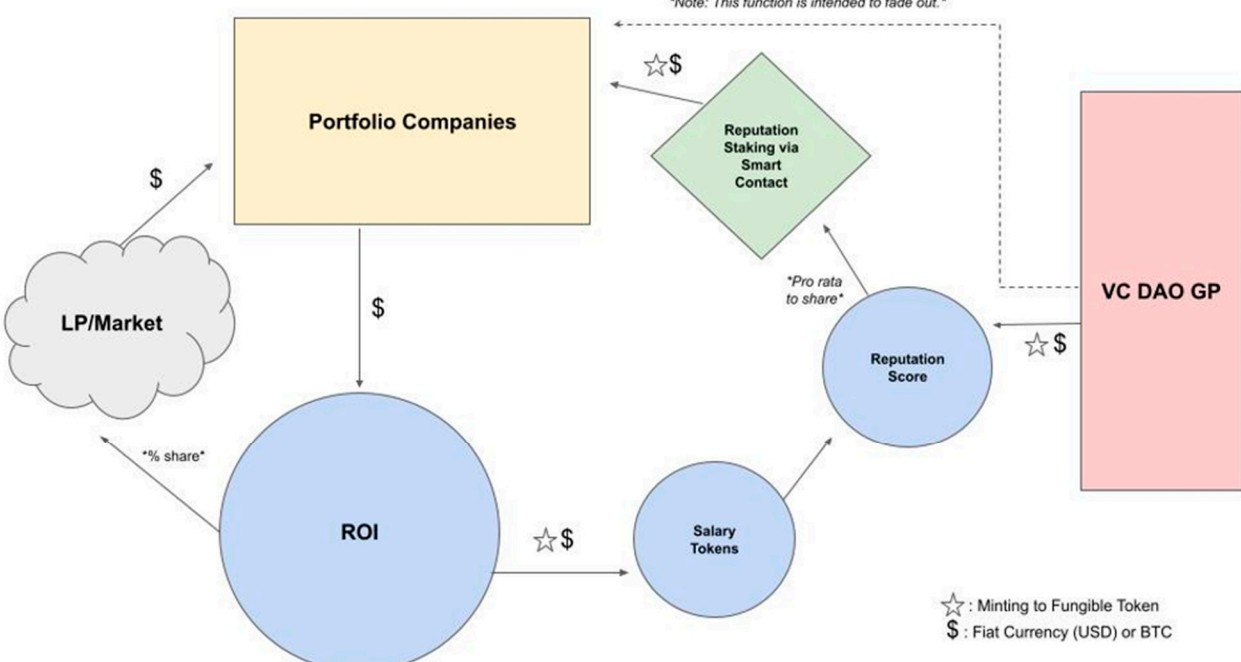

**Figure 1.** The incentive compatible model.

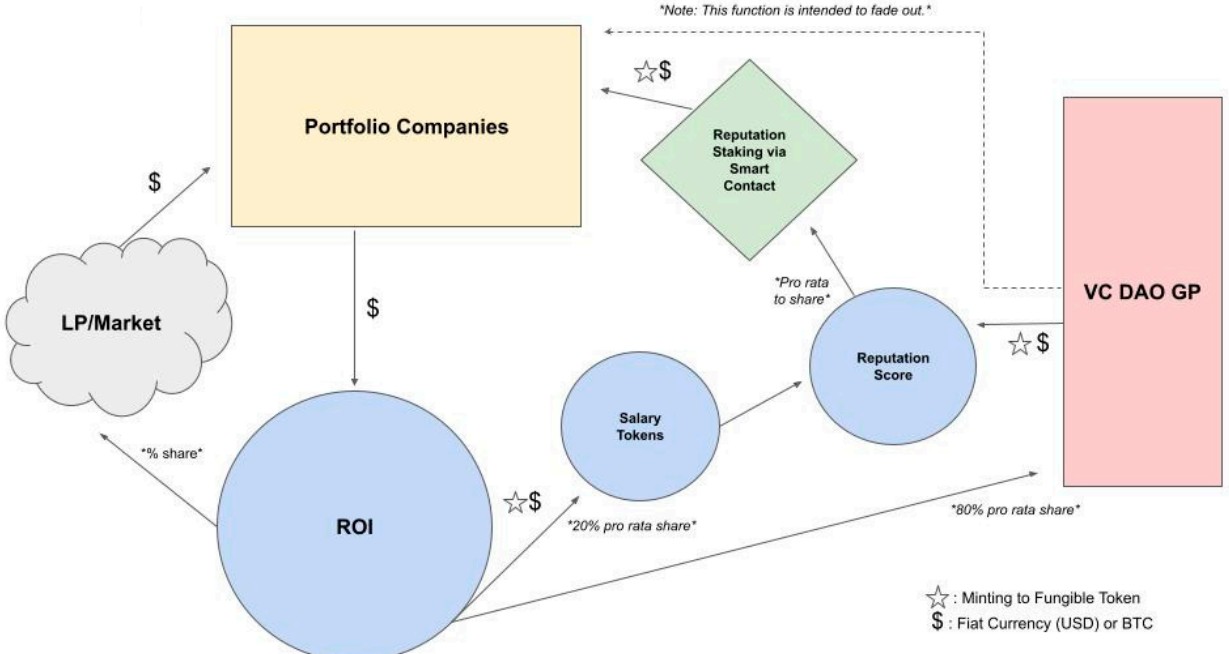

**Figure 2.** The hybrid model of a VC DAO.

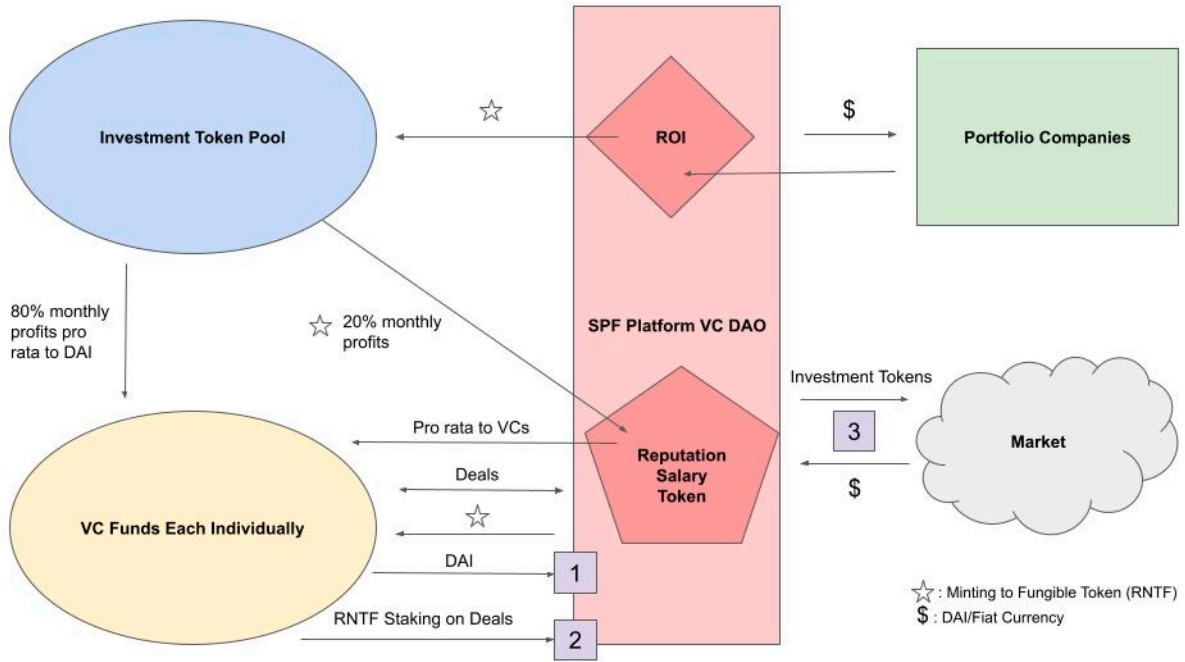

**Figure 3.** Sample legal structure with smart contracts.

### 5.1. Basic Model

This original model in Figure 1 illustrates how the signaling function of a well-functioning VC DAO can provide for the digital asset investment market. Here, the VC DAO GP fully and transparently allocates pooled assets to digital asset portfolio companies. An LP only participates by following the GP investments. The VC DAO GP does not benefit from those LP investments directly. As noted in the model, additional funding directly from the DAO.

Importantly, the model in Figure 1 does not rely on reputation staking exclusively. Rather, the VC DAO GP invests both fungible cryptocurrency from a pool as well as non-fungible and minted reputation on the portfolio companies. This duality does not allow for

the full benefits that are generated if a non-fungible reputation is staked alone on the deals. Over time, direct funding from the VC DAO GP will fall away. Subsequently, reputation scores feed into LP/market as LPs co-invest with GP reputation staked on proposed deals.

Figure 1, however, illustrates the early phases of a VC DAO GP that uses reputation as capital, as in this model, the ROI is already paid out in proportion to the reputation scores of each VC DAO GP.

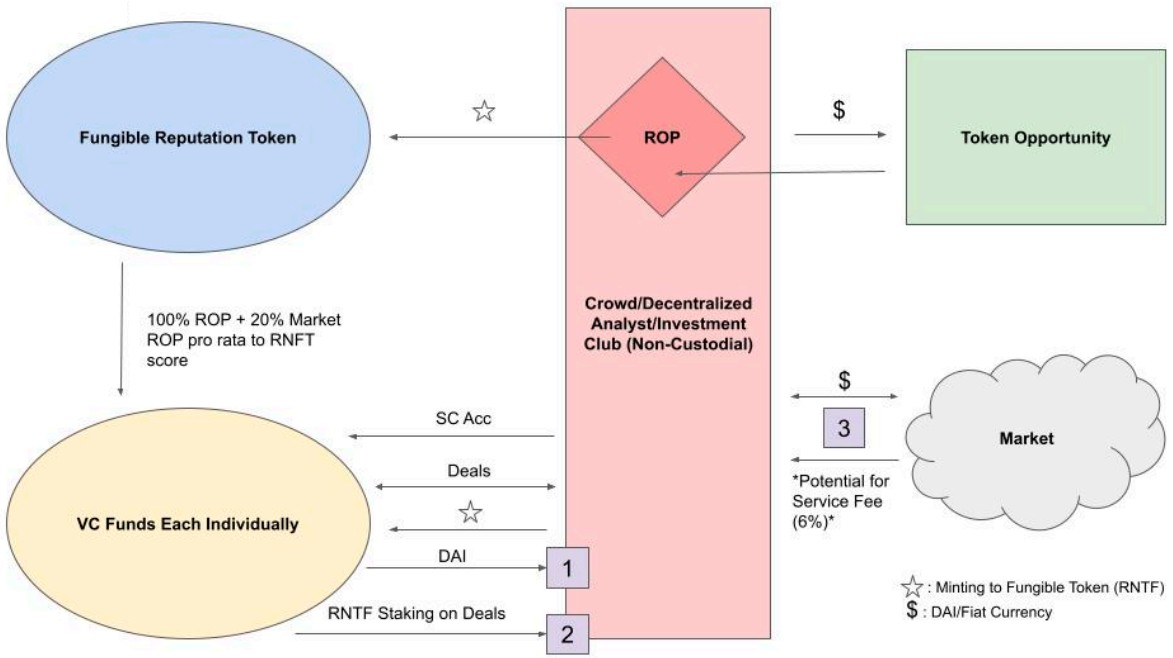

**Figure 4.** Non-Custodial DAO Investment Club Model with Smart Contracts.

### 5.2. Hybrid VC Model with Traditional Legal Structure

For many economic and business reasons and because of significant path dependencies in the capital formation and capital allocation process with existing venture capital models, the above illustrated ideal-typical reputation-based capital allocation model in Figure 1 is less likely to be created in the short term. Perhaps after a longer experimentation period with some of the hybrid models discussed below, the ideal-typical reputation-based capital allocation model will become more widely considered. Ultimately, all the models discussed in this article contribute to the much-needed experimentation with venture capital reputation models.

The below summary provides an overview of a hybrid model in Figure 2 that enables a traditional ROI for 80% of capital committed on a pro rata basis but also allows the use of reputation staking and staking rewards as salaries which are financed with 20% of the returns on capital committed; 80% of returns are allocated to investors pro rata as ROI and 20% of returns are allocated to the reputation salary pool.

GP whales get reputation tokens allocated pro rata to their capital commitments. Founders get 2% of AUM annually as a service fee via separate contract. Founders also get a percentage of reputation. GP whales initially fund projects and build reputation. Founders' marketing revolves around a 'ride with whales' mantra. LPs do not get reputation tokens but instead receive fungible investment tokens in the fund.

Investment tokens are the fungible instantiation of the reputation gains of the whales. As a track record forms, whales no longer have to stake capital for projects to which they stake reputation. LPs can bid for the right to commit capital for a project the whales upvoted via reputation stakes. Whales still participate in the success of each upvoted project through 20% of the fund's returns allocation to the reputation salary pool, which gets paid out proportional to each whale's reputation score.

Whales who proposed projects that were upvoted by the validation pool receive a bump up in their reputation. This process follows a process of decentralized governance (Calcaterra et al. 2018). Through the decentralized governance precedent system, projects that were upvoted by the whale validation pool have the highest comparative ROI, get more citations in the reputation system, and continually enhance the reputation of the whale who sourced and proposed it.

### 5.3. Hybrid VC Model with Smart Contracting

Figure 3 is available to show the sample legal structure with smart contracts. Internal and external relationships of the SPF are coordinated through smart contracts. VCs can either DAI invest as shown by number 1 or reputation stake shown by number 2. If the VC desires, over time, DAI investments directly from the VC to DAO would fall away and feed into the market function illustrated by the number 3. The market then co-invests based on the VC's RNFTb staking on deals.

Here, 20% ROI will be charged to the market for co-investing with the VCs (as the usual 2/20 deal) and then will fund the reputation salaries for the VC. Hence, in the model in Figure 3, VCs will fund their reputation salaries mostly through the public market.

In order to interact with the real world, the VC DAO needs a legal wrapper. Otherwise, everyone involved in the DAO may be jointly and severally liable in any jurisdiction. In the case of a VC DAO incorporated as a Swiss Association, the assembly (of members of the Swiss Association) would consist of all the VC funds that get upvoted as members (by the existing members following decentralized governance) after contributing capital. RNFT is minted and allocated to VCs in proportion to the DAI contributed by each VC. The Delegate Association Member (DAM) represents the VC DAO members in the Association and to the Association's board. The Association's board acts as the agent of the VC DAO in the real world.

The problem with this model is that VCs are incentivized partially to fund and stake only on the best deals and stake on the less optimal deals that are mostly funded by the market. This undermines their long-term reputation accumulation as some VCs may play favors and sacrifice their reputation to gain more on the investment side. It is possible to mitigate these incentives that undermine the long-term success of the system. For example, it is conceivable that staking on deals by VCs mandates capital commitments to the deals but that VCs can, over time, lower their capital commitments and increase their staking.

### 5.4. Incentive Optimized Reputation Model with Smart Contracts

A better system would avoid this scenario by emphasizing the reputation in the system by prohibiting capital investment into portfolio companies after an initial minting of reputation in proportion to incoming capital. For purposes of this minting ratio, and for the optimal incentivization thereafter, it would be preferable to keep the minting of reputation tokens equally proportional to incoming capital. In other words, in order to start every incoming venture capital fund with the same reputation capital, all incoming capital would preferably obtain the same pro rata value.

Figure 4 illustrates the early stages of a non-custodial DAO investment club model (DAOIC) with smart contracts. In this model, all of the ROP is minted into fungible reputation tokens that get paid as reputation salaries following decentralized governance. This model provides the best incentive alignment for DAOIC members with the highest potential return for all involved. Similar to Figure 3, VCs can either DAI invest as shown by number 1 or reputation stake shown by number 2. If the VC desires, over time, DAI investments directly from the VC to DAO would fall away and feed into the market function illustrated by the number 3. The market then co-invests based on the VC's RNFTb staking on deals. Note that in this figure, 20% of market earnings flow through fungible reputation tokens while the remaining 80% ROP is DAI invested by the market itself into the fund.

Every original member of the DAOIC starts with the same capital commitment that gets minted into RNFT. Future members may be onboarded with deal proposals by way of

the existing DAOIC member vote following decentralized governance. Newly onboarded DAOIC members start with the same capital commitment as the original members and the corresponding RNFT score.

As the public co-purchases with the DAOIC members and expects to pay a price for the right to benefit from the collective wisdom of the DAOIC members and the deal pipeline they can together generate, the public should expect to pay the usual 2/20 fees. Accordingly, the DAOIC members get paid 100% of their respective ROP from their capital commitment plus 20% ROP from the public's capital commitment on the respective deal.

Depending on the deal structure, the 20% ROP may significantly outweigh the 100% ROP each DAOIC member would receive from their own allocation of capital.

Over time, the members of the DAOIC do not need capital any longer. All they need is the ability to stake RNFTs on newly proposed incoming deals. The public market funds the deals, and the DAO members get paid via the 20% public ROP minting to fungible reputation token.

If members still wish to commit capital on particular deals, they can get into deals on the public market side, based on the same conditions. However, they would, in essence, be paying themselves with the 20% of ROP.

The removal of capital as a necessity for deal participation creates a very high level of flexibility for DAOIC members. It allows the DAOIC members to invest if they so choose but it does not force them to invest under any conditions. Capital calls are entirely unnecessary.

DAOIC members can decide which deals are so good that it would be worth paying themselves the 20% of ROP. Presumably, most deals would be worth it. However, the nature of the publicly listed fungible reputation token is a much better value proposition for the DAOIC members, especially in the long run. If a DAOIC member suffers a liquidity crisis, such a member still participates in the fungible reputation salary payouts.

Not only are members sharing in the ROP of the market with 20%, which may be much more in the aggregate than the individual DAOIC members' ROP, the model also amplifies the DAOIC member returns because only DAOIC members get paid in fungible reputation tokens.

Such tokens are themselves a significant value proposition because the fungible reputation token is based on the deal pipeline upvoted via RNFT. The token represents the collective wisdom of the DAO investment club members, and, in addition, the total ROP (100% Member PLUS 20% Public ROP) on each deal. The combined effect of these factors will likely cause the public perception and corresponding market valuation of the fungible reputation tokens to be very favorable.

The collective wisdom of DAOIC members helps hedge against purchase risk. Moreover, because the applicable decentralized governance with loosely and tightly coupled votes will very likely make all tightly coupled votes unanimous, no DAOIC member loses RNFTs and continues to receive the stream of fungible reputation tokens. Fungible reputation tokens can be sold to the market as needed by DAOIC members which provides enhanced liquidity for VC participants.

The life cycle of the DAOIC involves different phases. During phase 1, the DAOIC brings in members with their deals and mints RNFT in proportion to incoming capital. In phase 2, DAOIC members stake RNFT on incoming deals. The capital committed by DAOIC members is allocated on the deals and ROP is paid out in fungible reputation tokens in proportion to each DAOIC member's RNFT holdings. During phase 2, the fungible reputation token is listed on digital asset exchanges. This allows the DAOIC members to use the fungible reputation token for their own liquidity needs.

The shift away from capital starts incrementally. DAOIC members may still need to commit capital in phases 1 and 2. However, the move away from capital commitments is already possible. For example, in phases 1 and 2, if the market co-purchases at 50% of capital commitment of a given deal and 20% of the corresponding ROP on such a deal is allocated to the fungible reputation token minting pool, which is shared only by the

DAOIC members, then the DAOIC members would, in the aggregate, be better off keeping their own ROP at 100% (which is proportional to their capital commitment) combined with the 20% of ROP from public/market capital commitments in the fungible reputation token, which is paid out proportionally to RNFT score. This is especially true considering that the fungible reputation token is itself an investment proposition that amplifies the DAOIC members' initial capital commitment.

During phase 3, the public co-purchases deals based on the RNFT staking on deals by the DAOIC members. The public receives their ROP pro rata to capital commitment in DAI. Phase 3 enables a consolidation of the shift away from capital. In phase 3, if the market purchases 100% of capital commitment of a given deal, based on DAOIC member staking on the deal, and 20% of the corresponding ROP on such deal is allocated to the fungible reputation token minting pool, which is shared only by the DAO investment club members, then the DAOIC members would, in the aggregate, be better off keeping their own ROP at 100% (which is proportional to their capital commitment) combined with the 20% of ROP from public/market capital commitments in the fungible reputation token, which is paid out proportionally to RNFT score. DAOIC can still invest on the public side if they so choose.

## 6. Conclusions

The existing VC model is subject to significant ongoing downsides that inhibit its ability to provide long-term value to the tech startup community. The existing market for fiat VC investments is subject to several significant downsides, especially in comparison with the market for early round digital asset investments.

The reputation as capital model has the potential to overcome many of the downsides in the existing legacy VC market. Reputation as capital has the potential to lower capital requirements for VC businesses significantly and increase liquidity at unprecedented levels because VC fungible reputation tokens can be sold to the market as needed. In turn, the removal of capital as a necessity for deal participation creates a very high level of flexibility for VCs. It allows VCs to invest if they so choose but it does not force them to invest under any conditions. Capital calls become entirely unnecessary. VCs that suffer a liquidity crisis would still participate in the fungible reputation salary payouts.

The industry is still experimenting with different forms of DeFi capital replacement schemes. It will take time to filter out those designs that have unanticipated side effects, unexpected hidden incentives, and associated behavioral patterns and capital flows. Yet, the experimentation has already started in 2020 and new designs and projects materialize quickly with unexpected success stories.

**Funding:** This research received no external funding.

**Data Availability Statement:** Not applicable.

**Acknowledgments:** The author is grateful for many discussions with Craig Calcaterra on the uses of reputation systems. The author is grateful for outstanding research assistance from Hayley Howe and research librarian Elizabeth Hadden-Peck.

**Conflicts of Interest:** The authors declare no conflict of interest.

## Notes

1.   (Remeika et al. 2018, p. 1). For discussion of regulatory initiatives aimed to facilitate VC fundraising, see (Fenwick and Vermeulen 2019).
2.   (Fried and Hisrich 1994, p. 36). For discussion on agency theory and VC, see (Gompers 1995, pp. 1461, 1463–67).
3.   (Tyebjee and Bruno 1984, p. 1056). For in-depth discussion on syndication, see (Lerner 1994).
4.   (Tyebjee and Bruno 1984, pp. 1056–57). For discussion of the financing stage, see (Gompers 1995, pp. 1461, 1475–84).
5.   (Tyebjee and Bruno 1984, p. 1054). For discussion on VC firms taking companies public, see (Gompers 1996, p. 133).
6.   17 C.F.R. § 275.203(l)-1(a)(2).
7.   Cite to problems and prospects.

8    See (Bonini and Capizzi 2019) (analyzing data on Italian, German, UK and US VC markets); (Bonini et al. 2019; Löher 2016; Konrad 2019; Lin and Nestarcova 2019).

9    This section of text builds on established concepts discussed in the author's book entitled *Decentralization* co-authored by (Calcaterra and Kaal 2021).

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
