# Peer review of "Reputation as Capital—How Decentralized Autonomous Organizations Address Shortcomings in the Venture Capital Market"

_jrfm, doi:10.3390/jrfm16050263_

Round 1

Reviewer 1 Report

I must congratulate author(s) for their work, and the capacity to present such an interesting and recent subject. I am sure this subject worth a very interesting paper.

The presentation is very clear and focused, and the point that author(s) want to stress is well explained.

I have serious concerns, however, concerning the scientific soundness of the paper, mostly because there are several important sections on the theme of the literature review that are not adequately supported (from my point of view) from recent scientific literature, as it should (that is the case, at least, of sections 2.1.a); 2.2.; 3.; and 4.). Moreover, where there is some support, authors seem to depend heavily on one source/article, like Bonini or Tyebjee, thus impoverishing the results of what could be very interesting research.

The same happens with the Section of the "Model Evolution" (Section 5) where we cannot ascertain whether the proposed figures are from the research of the authors or are taken from other authors.

These are some subjects that are crucial, from my point of view, to guarantee the soundness of this research and to give it the importance that I consider it might have.

These are also very important topics to support the soundness of any paper published on JRFM. I am sure these are issues that, with additional work, authors could overcome easily, enriching their research, and making a concrete important contribution to this subject.

I very much look forward for the future version of the paper :).  

Author Response

Response:

To address the concerns laid out by Reviewer 1, we have added numerous sources to the paper throughout as well as diversifying existing sources. Specifically we have added sources to Sections 1-3. I also made sure to find and add some more current sources. I understand your concern with some sources being relied upon too heavily and hopefully that issue has been rectified. I added several comments throughout section 5 to address your concern regarding ownership of the graphs. To be clear, they are all original works by the author. Finally, small grammatical/stylistic edits were made to the paper.

Reviewer 2 Report

In this paper, the author discussed the problems of the current VC market and proposed that decentralized reputation governance models could optimize the market. The contribution of this paper is, however, unclear to me. Section two of the paper discusses the shortcomings of the existing VC model, while sections three through five describe alternative models without providing convincing evidence. A different organization and description of the study may be considered by the author. As of now, I am unsure of what it contributes to the literature. 

Author Response

Response: To address this concern, we have added more support to sections 3-5 in order to help support section 2’s assertions. The contribution of this paper is to compare funding mechanisms between a traditional VC and other methods specifically from an underwriting perspective. This is valuable for readers on both sides of a transaction. Furthermore, it offers a more broad perspective on the proposed mechanism from an industry wide perspective. Grammatical/stylistic edits were made to the paper as well.

Reviewer 3 Report

This appears to me as review article. I do not see clearly the contribution of the author. The topic is quite an interesting topic and will be of interest to many. 

Author Response

Response: The contribution of this article lies within its application of alternative funding mechanisms and the modeling which addresses how they interact with both the market and VC firms. Grammatical/stylistic edits were made to the paper as well.

Reviewer 4 Report

I would like to thank author for his work about the interesting and timely topic of the paper. Overall, I think the paper is good. However, I found that the overall quality of the paper still needs some improvement before the publication. Please find my constructive comments below.

1)    First of all, the current format of the paper needs to be revised.

2)    The abstract also needs to be more elaborated. The present version is simply too short. Please consider revising it.

3)    Author should clearly provide the main contributions of the article presenting the model of reputation as capital.

4)    More background about the proposed models presented in section 5 should be provided in the earlier section to help readers better understand the working mechanisms behind the models.

5)    In section 5, the 4-model evaluation, it may be better to compare in a table format these 4 models and their advantages and shortcomings. Also, I don’t get the number [1], [2], and [3] that appeared in Figure 1-4, please explain what they are for.

6)    Limitations of the proposed model should be provided, as well as future research direction.

Author Response

Response: We have revised the current submission with all of your comments in mind. Specifically, we added to section 5 and re-worked the models to address reader confusion.

  • The format was revised and is in good shape now.
  • We have added language to the abstract to address your concern.
  • Author contribution has been clarified. See footnotes in sections 4 and 5.
  • We feel this material would not make sense to discuss earlier in the paper. However, we did add some more clarifying language to the text addressing each image in an effort to make it more comprehensive.
  • We have reworked the images to make more sense for the reader. This should help with some of the confusion. We also worked on better labeling within the graphs.
  • Future research direction and limitations will not be added at this time.

Round 2

Reviewer 1 Report

Dear author,

I congratulate you for your paper's proposal. I believe it deals with a veryh interesting and important topic, is very actual and your idea seems to be really interesting.

Despite these considerations, after reading carefully your paper, I consider that it needs some crucial changes in order to be considered adequate for publication. I explain below my ideas:

1-   I like very much the way you write. However, we must not forget that JRFM is a scientific based journal. As so, your presentation must follow the scientific method. This means that your ideas must be supported either in statistical data, and/or in other scientific literature on the topics discussed. I notice many parts of your article where it seems that you are presenting your personal ideas (and I must say that, from an empirical point of view, I agree with most of your ideas), without these statistical or theoretical support. This must be changed;

2-   I consider that author must revise the objectives of the paper, and follow a concrete line of thought in order to reach those objectives. Sometimes the narrative used is not very clear where we are heading or what are we trying to reach…

3-   Literature review is critical in any scientific paper. This literature must be based in peer-reviewed journals, and sometimes in very well known books on the topic. I consider that the section of literature review needs a clear revision in order to reach this stage. It is not usual to use the same reference so many times as the authors uses many of them. This must be changed;

4-   Following the previous thought, bibliographic references seem to be wrong, since they are remitting to previous references that are not correct;

5-   In the same line of thought, the methodology of the paper must be revised. What do we intend to reach with this paper. How are we going to reach it? Models are presented, but we cannot ascertain the theoretical base used to reach those models.

6-   Moreover, there are a bunch of acronyms used that are not adequately explained. It is not correct to make readers investigate what authors want to say by using the proposed acronyms…

7-   Also, if we are comparing models, we must be informed of the criteria used to compare them. This, from my point of view, doesn’t happen in this paper;

8-   Finally, if the previous ideas and the consequent recommendations are implemented, I believe that the conclusions will be more clear and reinforced regarding what is expected. The poor conclusions presented are a consequence of the inadequate objectives defined for the paper. I am sure that if the author follows these recommendations, he will get much stronger support for the results expected, and then for the conclusions presented.

Having said this, I must reinforce the idea that I believe there is a strong potential in this paper, and that the idea that author proposed can be much supported if the above mentioned recommendations are followed.

Author Response

1-   I like very much the way you write. However, we must not forget that JRFM is a scientific based journal. As so, your presentation must follow the scientific method. This means that your ideas must be supported either in statistical data, and/or in other scientific literature on the topics discussed. I notice many parts of your article where it seems that you are presenting your personal ideas (and I must say that, from an empirical point of view, I agree with most of your ideas), without these statistical or theoretical support. This must be changed;  

I AM HAPPY TO ADD STATISTICAL SUPPORT IF THERE IS ANY. I AM NOT AWARE OF ANY RESEARCH THAT SUPPORTS THE POINTS I AM MAKING. PLEASE FEEL FREE TO ADD THE SOURCES YOU REQUIRE AND I WILL HAPPILY CONSIDER THEM

2-   I consider that author must revise the objectives of the paper, and follow a concrete line of thought in order to reach those objectives. Sometimes the narrative used is not very clear where we are heading or what are we trying to reach…

THE LINE OF ARGUMENT IS VERY CLEAR, REPUTATION REPLACES CAPITAL OVER TIME AND CHANGES HOW VENTURE CAPITAL WORKS WITH A BETTER MODEL THAT IS CHEAPER FOR ALL INVOLVED 

3-   Literature review is critical in any scientific paper. This literature must be based in peer-reviewed journals, and sometimes in very well known books on the topic. I consider that the section of literature review needs a clear revision in order to reach this stage. It is not usual to use the same reference so many times as the authors uses many of them. This must be changed;

HAPPY TO REMOVE THE MULTIPLE CITATIONS TO THE SAME SOURCE - PLEASE FEEL FREE TO REMOVE WHERE NEEDED. I AM NOT AWARE OF OTHER LITERATURE IN THIS REGARD AND HAVE THEREFORE LIMITED CITATION TO THE LITERATURE AVAILABLE. IF YOU CONSIDER OTHER SOURCES THAT YOU WISH TO CITE, I WILL BE HAPPY TO USE THEM IF THEY ARE APPROPRIATE. 

4-   Following the previous thought, bibliographic references seem to be wrong, since they are remitting to previous references that are not correct;

I AM HAPPY TO HAVE MY LIBRARY STAFF FIX THIS. PLEASE POINT ME TO THE SOURCES YOU DEEM WRONG AND WE WILL ADDRESS THEM 

5-   In the same line of thought, the methodology of the paper must be revised. What do we intend to reach with this paper. How are we going to reach it? Models are presented, but we cannot ascertain the theoretical base used to reach those models.

THIS IS A NEW FIELD AND THE COMMENTER SEEMS TO UNDERMINE THE PROLIFERATION OF THE NEW FIELD WITH THIS GENERIC COMMENT. 

6-   Moreover, there are a bunch of acronyms used that are not adequately explained. It is not correct to make readers investigate what authors want to say by using the proposed acronyms…

PLEASE POINT THEM OUT IN THE MANUSCRIPT AND I WILL DO MY BEST TO EXPLAIN THEM WITH A LEGEND. THIS IS AGAIN AN NEW FIELD AND NEW RESEARCH WITH NEW IDEAS THAT OFTEN IS NOT INTUITIVE TO THE LAY AUDIENCE. 

7-   Also, if we are comparing models, we must be informed of the criteria used to compare them. This, from my point of view, doesn’t happen in this paper;

NOT SURE WHAT THIS MEANS - THE MODELS ALL BUILD ON EACH OTHER AND THE COMPARISON STARTS WITH THE FIRST MODEL 

8-   Finally, if the previous ideas and the consequent recommendations are implemented, I believe that the conclusions will be more clear and reinforced regarding what is expected. The poor conclusions presented are a consequence of the inadequate objectives defined for the paper. I am sure that if the author follows these recommendations, he will get much stronger support for the results expected, and then for the conclusions presented.

PLESE ADVISE ON NEXT STEPS AFTER GIVING MORE CLARIFICATION AS REQUESTED. 

Reviewer 2 Report

I sincerely appreciate the additional sections the authors included in the updated version. This study provides a broader view of the proposed mechanism, which will contribute to the literature. 

Author Response

I appreciate your advice

Reviewer 3 Report

I am fine with the revised version.

Author Response

I appreciate your advice